# A Hybrid Perovskite-Based Electromagnetic Wave Absorber with Enhanced Conduction Loss and Interfacial Polarization through Carbon Sphere Embedding

**DOI:** 10.3390/nano14191566

**Published:** 2024-09-27

**Authors:** Xuehua Lian, Yao Yao, Ziming Xiong, Yantao Duan, Jianbao Wang, Shangchen Fu, Yinsuo Dai, Wenke Zhou, Zhi Zhang

**Affiliations:** 1Field Engineering College, Army Engineering University of PLA, Nanjing 210007, China; lianxuehua@outlook.com; 2State Key Laboratory for Disaster Prevention & Mitigation of Explosion & Impact, Army Engineering University of PLA, Nanjing 210007, China; yylgd1126@outlook.com (Y.Y.); xzm992311@163.com (Z.X.); 3Electromagnetic Environmental Effects Laboratory, Army Engineering University of PLA, Nanjing 210007, China; dcmchdyt@126.com (Y.D.); zwang0417@outlook.com (J.W.); fshangchen@sina.com (S.F.); 4Position Engineering Research Office, Army Engineering University of PLA, Nanjing 210007, China; ysdaii@aeu.edu.cn

**Keywords:** hybrid perovskite, carbon sphere, conduction loss, interfacial polarization, electromagnetic wave absorption

## Abstract

Electronic equipment brings great convenience to daily life but also causes a lot of electromagnetic radiation pollution. Therefore, there is an urgent demand for electromagnetic wave-absorbing materials with a low thickness, wide bandwidth, and strong absorption. This work obtained a high-performance electromagnetic wave absorption system by adding conductive carbon spheres (CSs) to the CH_3_NH_3_PbI_3_ (MAPbI_3_) absorber. In this system, MAPbI_3_, with strong dipole and relaxation polarization, acts dominant to the wave absorber. The carbon spheres provide a free electron transport channel between MAPbI_3_ lattices and constructs interfacial polarization loss in MAPbI_3_/CS. By regulating the content of CSs, we speculate that this increased effective absorption bandwidth and reflection loss intensity are attributed to the conductive channel of the carbon sphere and the interfacial polarization. As a result, when the mass ratio of the carbon sphere is 7.7%, the reflection loss intensity of MAPbI_3_/CS reaches −54 dB at 12 GHz, the corresponding effective absorption bandwidth is 4 GHz (10.24–14.24 GHz), and the absorber thickness is 2.96 mm. This work proves that enhancing conduction loss and interfacial polarization loss is an effective strategy for regulating the properties of dielectric loss-type absorbing materials. It also indicates that organic-inorganic hybrid perovskites have great potential in the field of electromagnetic wave absorption.

## 1. Introduction

Studying new electromagnetic absorbing materials with effective absorption bandwidths is urgent to cope with the electromagnetic interference and radiation caused by electronic equipment [1,2,3,4,5,6]. In recent years, organic-inorganic hybrid perovskite has been proven to be a potential wave-absorber because of its high element tolerance, changeable lattice structure, and mild solution preparation methods [7,8,9,10,11]. However, problems such as relatively low absorption intensity (−3.3 dB) and large thickness of the absorber (4 mm) limit the in-depth application of hybrid perovskites in the field of electromagnetic wave absorption [8]. It has been found that preparing multi-component composites, enhancing polarization, and constructing multiple loss mechanisms are important ways of adjusting the impedance matching of absorbing materials [12,13,14,15,16,17,18,19,20,21,22]. Therefore, how to design a perovskite-based absorbing system with a precisely tunable composition and structure to improve reflection loss and absorption bandwidth is an urgent problem to be solved.

Meanwhile, the carbon sphere (CS) has become an ideal wave-shielding matrix due to its high conductivity, low density, and stable structure [23,24]. The surficial porous structure of the CS can reflect and scatter the incident electromagnetic wave multiple times, enhancing the attenuation of incident electromagnetic waves. Ji et al. prepared mesoporous CSs by hard template etching, adjusted the mesoporous size, and obtained a maximum reflection loss of −50.9 dB [25]; Han et al. reported a Mo_2_C@C nanosphere. By utilizing multiple loss mechanisms such as conductivity loss, dipole polarization loss, and interfacial polarization loss, the reflection loss at 12.5 GHz reaches −48.0 dB, and the effective absorption bandwidth is 3.5–18.0 GHz [26]. Therefore, through combining CSs with other wave-absorbing materials, the polarization loss intensity of the wave-absorbing system can be improved by taking advantage of its characteristics, such as the adjustable shape and size and the large specific surface area of CSs. Meanwhile, CSs can form conductive networks in the composite wave-absorbing system, increasing conductivity loss intensity.

This work uses CH_3_NH_3_PbI_3_ (MAPbI_3_) as the host to composite porous CSs to construct MAPbI_3_/CS. We obtained an optimal reflection loss intensity of −54 dB and an effective absorption bandwidth of 4 GHz (10.24–14.24 GHz) when the absorbent’s additive amount was 40 wt% and the mass ratio of MAPbI_3_:CS was 12:1. Further analyzing the internal polarization relaxation process of composites, we found that the introduction of CSs enhances the interfacial polarization and impedance matching of the system. As a result, it improved the intensity of polarization loss. At the same time, the high conductivity of the CSs improved the dielectric loss intensity of the system. Through this work, we explored the influence of the content of CSs on the absorbing properties of a MAPbI_3_/CS composite, which laid a foundation for the design and preparation of high-performance perovskite-based absorbing materials.

## 2. Materials and Methods

### 2.1. Synthesis of MAPbI_3_ Microcrystals and Carbon Sphere

MAPbI_3_ was synthesized by an anti-solvent method [8]. The CS material was obtained by carbonizing phenolic resin balls with phenolic resin balls as precursors. Phenolic resin spheres were prepared by suspension polymerization. First, 0.9 g of 3-aminophenol was fully dissolved in a mixed solution of deionized water and ethanol, and the deionized water and ethanol solutions were of 60 mL and 25 mL, respectively. Then, 0.27 mL of ammonia water was added dropwise to the mixed solution under stirring, and we continued to add 1.215 mL of formaldehyde solution dropwise after the solution was stable. The mixed solution was continuously stirred for 4 h, and then the mixed solution was centrifuged and washed several times with deionized water and ethanol, and the obtained milky white material was the phenolic resin ball precursor. Finally, the dried phenolic resin ball precursor was put into a tubular high-temperature heating furnace for annealing treatment. The annealing temperature was 800 °C, the holding time was 3 h, and argon was used as the atmosphere condition. The black powder after the annealing treatment was the CSs.

### 2.2. Synthesis of MAPbI_3_/CS Composites

A CS suspension was prepared by dispersing CS powder in γ-butyrolactone solution at a concentration of 30 mg/mL, and then stirring at 25 °C for 3 h. Then, the MAPbI_3_ precursor solution and the CS suspension were mixed according to a certain proportion and stirred at 25 °C for 1 h. According to the mass ratios of 24:1, 16:1, 12:1, 8:1, and 6:1 (MAPbI_3_ versus CSs), the MAPbI_3_/CS composites were prepared and marked as MS-1, MS-2, MS-3, MS-4, and MS-5, respectively. After the reaction, the MAPbI_3_/CS mixed solution was quickly added dropwise into the excess anisole solution. A large amount of black suspended solids could be observed during the reaction. Then, the anisole solution was placed under a nitrogen atmosphere for 12 h. Finally, the lower precipitate in anisole solution was washed by centrifugation with isopropanol and n-hexane several times, and the resulting product was dried under vacuum at 80 °C for 24 h. The obtained black sample was the MAPbI_3_/CS composite.

### 2.3. Structural Characterization

XRD analysis was carried out using a D8 Advance X-ray diffractometer. The operating voltage of the instrument was 50 KV, the operating current was 100 mA, and the Cu target Kα ray (λ=1.54187 Å) test range was 2θ = 5–80°. Raman analysis was carried out using an Acutech Scientific 1500 RAMAN spectrometer (Arcadia, CA, USA). Photoluminescence (PL) analysis was conducted by FLS-1000, and the wavelength range of the samples was 200–800 nm. SEM analysis was carried out using the Nova NanoSEM 450 instrument (FEI, Eindhoven, The Netherlands).

### 2.4. Material Microwave Absorption Characterization

The electromagnetic parameters were tested via a PNA N5244A vector network analyzer using the coaxial ring method. The frequency range of the test electromagnetic wave was 2–18 GHz, and the step size was 0.02 GHz. Then, the sample/paraffin mixture in a mass ratio of 2:3 was pressed into a ring specimen with an outer diameter of 7 mm and an inner diameter of 3.04 mm using a mold. Finally, the coaxial ring was put into the vector network analyzer for testing [27,28].

## 3. Results and Discussion

To investigate the physicochemical properties of MAPbI_3_/CS crystals, such as composition and lattice structure, we employed X-ray diffraction (XRD), Raman spectra, and Photoluminescence (PL) spectra (Figure 1). Figure 1a shows the XRD patterns of the MAPbI_3_/CS, MAPbI_3_, and CSs. It can be seen that the characteristic diffraction peaks in MAPbI_3_/CS (MS-1, MS-3, MS-5) are consistent with the main diffraction peaks of tetragonal-phase MAPbI_3_ crystals [29,30]. In addition, with the increased content of CSs, the intensity of the diffraction peak at 2θ = 12.6°, corresponding to PbI_2_, increased. The emergence of PbI_2_ indicates that introducing CSs could induce the partial decomposition of MAPbI_3_. In addition, it can be found that the characteristic diffraction peak of the CSs appears near 2θ = 24.1°, and the full wide of half maximum (FWHM) is relatively large, indicating that the CSs are partially graphitized.

Further, we contrasted the Raman spectra of the MAPbI_3_, CSs, and MAPbI_3_/CS (Figure 1b). By calculating the intensity ratio of the D_1_ and G_1_ peaks in Figure 1b, it is found that the I_D1_/I_G1_ values of MAPbI_3_, MS-1, MS-3, and MS-5 are 0.61, 0.71, 0.85, and 0.87, respectively. It can be found that with the increased content of CSs, the ID_1_/IG_1_ value of MAPbI_3_/CS increases. In the published works, a higher ID_1_/IG_1_ value indicates higher defect densities, which may originate from space charge polarization [31,32,33]. Thus, the CSs could impact the nucleation and crystallization process of MAPbI_3_ and induce defects. Although introducing CSs causes the partial decomposition of MAPbI_3_, the generated defects and vacancies may be conducive to electromagnetic wave absorption because the increased defects will cause increased space charge polarization, which may enhance the intensity of the polarization loss. Further, the D_2_ and G_2_ peaks correspond to the crystal vibration of carbon in CS at the symmetry point of A_1g_ and the first-order scattering of E_2g_ phonons by carbon atoms in the sp^2^ orbital, respectively [34]. As shown in Figure 1b, for each content of CSs, the ID_2_/IG_2_ values of MAPbI_3_/CS are 0.98, which indicates that CSs can stably exist.

According to Figure 1c, with the increased content of CSs in MAPbI_3_/CS, the PL peak position belonging to MAPbI_3_ (770 nm) showed a slight blue shift of about 5 nm, and the FWHM became larger. A larger FWHM indicates the partial decomposition of MPI crystals, which is consistent with the diffraction peak results of PbI_2_ in the XRD pattern. In addition, electrical conductivity is an important index for evaluating the conduction loss of the composites. According to Figure 1d, the conductivity of the MAPbI_3_ crystal is 0.2 S m^−1^. With the addition of CSs, the conductivity of MAPbI_3_/CS significantly improved, reaching the highest 4.3 S m^−1^ for MS-3. So, introducing CSs can improve the movement of free electrons between MAPbI_3_ crystals. However, the conductivity of MS-5 (3.9 S m^−1^) is lower than that of MS-3. The decreased conductivity can be explained by the fact that excessive CSs will lead to the decomposition of MAPbI_3_ crystals, reducing the entire conductivity of MAPbI_3_/CS.

The micro-morphology of MAPbI_3_/CS under different contents of CSs was characterized by SEM (Figure 2), in which MAPbI_3_ were bulk crystals [35] and CSs are typical spheres [36], with a radius of 0.5 μm, in all groups of components (MS-1, MS-3, MS-5). With the increased content of CSs, the size of the MAPbI_3_ crystals gradually decreased from larger than 5 μm (Figure 2a) to less than 1 μm (Figure 2c), which verified the above inference that the nucleation of the MAPbI_3_ crystals was affected by the CSs. At the same time, when the CS content is relatively low, CSs are sporadically distributed on the surface of the MAPbI_3_ crystals (Figure 2a). When the content of CSs increases, the CSs agglomerate (Figure 2c). In these scenarios of Figure 2a,c, the CSs cannot effectively improve the conductivity and interfacial polarization of the system. Only when the CS content is appropriate (Figure 2b) could CSs uniformly distribute on the surface or in the grain boundaries of MAPbI_3_, constructing a conductive network between MAPbI_3_ grains and maximizing the intensity of interfacial polarization in the system.

The complex permittivity and complex permeability of the MAPbI_3_ crystals and CSs were characterized, as shown in Figure 3. For the complex permittivity, the real part (ε′) and the imaginary part (ε″) of the CSs are much larger than the ε′ and ε″ of MAPbI_3_. Based on ε″ ≈ 1/πε_0_ρf [37], the conductivity of MAPbI_3_/CS mainly originates from the CSs. For the complex permeability, it can be seen that in the range of 2–18 GHz, the real part (μ′) and the imaginary part (μ″) of the CSs are larger than the μ′ and μ″ of MAPbI_3_. Furthermore, the μ′ and μ″ of the CSs are around 1.0 and 0.05, respectively. Therefore, it can be judged that MAPbI_3_/CS composites may be dominated by dielectric loss, not magnetic loss.

Then, the complex permittivity and complex permeability of MAPbI_3_/CS are shown in Figure 4. The ε′ and ε″ of all the MAPbI_3_/CS samples are higher than the MAPbI_3_ crystals, which is consistent with the results of the conductivity test. For the MS-1 and MS-2, the data of ε′ match the law of the resonant dispersion curve (Lorentz model) [38,39]. The peaks of ε″ appear at 12.0 GHz and 14.0 GHz, indicating that the electronic transition generates resonant absorption of electromagnetic waves. Therefore, the MS-1 and MS-2 samples may appear to have strong electromagnetic wave absorption at 12.0 GHz and 14.0 GHz, respectively. The ε′ and ε″ of MS-3 present multiple peaks within 10–16 GHz, indicating that the MS-3 may exhibit obvious polarization relaxation and resonance absorption, which may include (1) resonance absorption generated by electronic transitions, (2) dipole polarization of MA^+^ and space charge polarization caused by defects, and (3) turning-direction polarization of an intrinsic dipole moment [8]. Therefore, it can be predicted that the MS-3 may appear to have strong wave-absorption capabilities within 10–16 GHz. The μ′ and μ″ of MAPbI_3_/CS are low, indicating that MAPbI_3_/CS has poor magnetic loss to electromagnetic waves. The peaks of μ′ and μ″ in MS-1 and MS-2 may originate from eddy current loss, natural resonance, and exchange resonance. Further analysis of the coefficient of eddy current loss is necessary.

To evaluate the dissipation ability of the material to the incident electromagnetic wave, we calculate the dielectric loss tangent and magnetic loss tangent [40]. The tanδ_ε_ and tanδ_µ_ of MAPbI_3_/CS are shown in Figure 5a. Within 10–16 GHz, the peak positions of tanδ_ε_ and tanδ_µ_ of MS-1, MS-2, and MS-3 are consistent with ε″ and μ″, respectively. Then, we calculate the eddy current loss coefficient (C_0_) of MAPbI_3_/CS at each content of CS [41]:(1)$$C_{0}=\frac{\mu^{\prime \prime }}{\sqrt{\mu^{\prime }}f}$$

As shown in Figure 5b, within 2–10 GHz, the values of C_0_ fluctuate significantly within the electromagnetic frequency, indicating that eddy current loss does not exist [41]. Within 14.5–18.0 GHz, the value of C_0_ is close to 0, indicating obvious eddy current loss. However, the tanδ_µ_ of MAPbI_3_/CS is close to 0 within 14.5–18.0 GHz. So, the magnetic loss intensity of MAPbI_3_/CS is negligible, and the electromagnetic wave absorption is dominated by dielectric loss. This slight magnetic loss may originate from the weak natural lattice resonance.

Furthermore, the attenuation constant (α) is calculated according to the following Formula [42]:(2)$$\alpha =\frac{\sqrt{2}\pi f}{c}\times \sqrt{\left(\mu^{\prime \prime }\varepsilon^{\prime \prime }\;-\;\mu^{\prime }\varepsilon^{\prime }\right)\;+\;\sqrt{{\left(\mu^{\prime \prime }\varepsilon^{\prime \prime }\;-\;\mu^{\prime }\varepsilon^{\prime }\right)}^{2}\;+\;{\left(\mu^{\prime }\varepsilon^{\prime \prime }\;+\;\mu^{\prime \prime }\varepsilon^{\prime }\right)}^{2}}}$$
where c is the speed of light in vacuum. Z can be calculated according to the following Formulas [43]:(3)$$Z=Z_{r}/Z_{0}$$
(4)$$Z_{r}=Z_{0}\sqrt{\left|\mu_{r}\right|/\left|\varepsilon_{r}\right|}$$
where Z_r_ is the material’s intrinsic impedance and Z_0_ is the free space impedance. As shown in Figure 5c, the α of MS-1, MS-2, and MS-3 is larger than 100 within 11.0–15.0 GHz, although MS-1 and MS-2 exhibit a jitter at 12 GHz and 14 GHz, in comparison with MS-3. However, MS-3 shows a larger α within 2–18 GHz, except for the peaks at 12 GHz and 14 GHz. As shown in Figure 5d, all the intrinsic impedance ratios of MAPbI_3_/CS at each content of CS exceed 0.3 within 2–18 GHz. Therefore, all the MAPbI_3_/CS composites have excellent transmittance to electromagnetic waves [44], and the component of CS in MS-3 is optimum. Meanwhile, we investigate the Debye relaxation properties of MAPbI_3_/CS, which can be found in the Appendix A.

The reflection loss (RL) of MAPbI_3_/CS in the range of 2–18 GHz can be calculated according to the following Formulas [25,45]:(5)$$Z_{\mathrm{in}}=Z_{0}\sqrt{\frac{\mu_{r}}{\varepsilon_{r}}}\tan h\left(j\frac{2\pi \mathrm{fd}}{c}\sqrt{\mu_{r}\varepsilon_{r}}\right)$$
(6)$$\mathrm{RL}\left(\mathrm{dB}\right)=20\mathrm{lg}\left|\frac{Z_{\mathrm{in}}-Z_{0}}{Z_{\mathrm{in}}+Z_{0}}\right|$$
where Z_in_ is the input impedance. Figure 6 exhibits RL values for MAPbI_3_, CS, and MAPbI_3_/CS. For MS-3, the effective electromagnetic wave absorption bandwidth (EAB) reaches 4 GHz (10.24–14.24 GHz), with a maximum reflection loss of −54 dB at 12 GHz and a thickness of 2.9 mm. In contrast, MS-1 and MS-2 only exhibit reflection loss around −42 dB at 12 GHz and 14 GHz with just 1 GHz of EAB. For MS-4 and MS-5, neither of the RLs reach −10 dB within 2–18 GHz. All the variation laws of reflection loss of MS under each CS ratio are consistent with the corresponding electromagnetic parameters, attenuation loss coefficient, and intrinsic impedance ratio in Figure 4 and Figure 5. For MAPbI_3_/CS, the wave dissipation contains polarization loss from MAPbI_3_ and conduction loss from CSs. For MAPbI_3_, the MA^+^ dipole, crystal defects, and crystal interface are the origins of the polarization loss. Combining the above data, we can demonstrate that induced CSs can enhance the intensity of the polarization loss and conduction loss. However, excessive CSs can cause the severe decomposition of MAPbI_3_ crystals, weakening the dipole and interfacial polarization loss.

According to the reflection loss performance of MAPbI_3_/CS in Figure 6, we extracted the EAB and its corresponding thickness, the maximum reflection loss intensity, and the corresponding thickness of MS-1, MS-2, and MS-3, as shown in Figure 7a,b. The MS-3 has the highest EAB and maximum reflection loss intensity, which are the optimum performance under 40 wt% addition. In addition, the maximum reflection loss intensity, normalized input impedance, and matched thickness correlations of the MS-1, MS-2, and MS-3 samples are shown in Figure 7c–e, where the normalized input impedance is calculated according to Formula (7) and the matched thickness is calculated according to the following Formula [37]:(7)$$t_{m}=\frac{n\lambda }{4}=\frac{\mathrm{nc}}{\left(4f_{m}\sqrt{\left|\mu_{r}\right|\left|\varepsilon_{r}\right|}\right)},\;\left(n=1,3,5,\ldots \right)$$

When the normalized input impedance values of the MS-1, MS-2, and MS-3 samples are close to 1.0, the reflection loss intensity of the samples reaches its maximum value. To summarize, MAPbI_3_/CS is a dielectric-loss-dominated absorbing material in which MAPbI_3_ crystals contribute to the core of dipole and space charge polarization. CSs enhance the interfacial polarization of the system and improve the movement of free electrons in MAPbI_3_ crystals due to their high surface area and high conductivity.

## 4. Conclusions

A MAPbI_3_-dominated MAPbI_3_/CS composite with a simple synthesis method and improved electromagnetic wave absorption ability was successfully developed. MS composites were characterized and analyzed by various means, and the electromagnetic wave absorption properties were deeply studied. We explored the optimal ratio of CS components in MAPbI_3_ and analyzed the wave-absorption mechanism. When the mass ratio of MAPbI_3_:CS is 12:1, the composite absorber reaches a high absorption strength of −54 dB at 12 GHz (40 wt%) and an effective absorption bandwidth of 4 GHz, revealing a promising application foreground of perovskite materials in the field of electromagnetic wave absorption. Moreover, this work proves the possibility of preparing electromagnetic wave-absorbing materials by combining perovskite-based material with multi-dimensional carbon materials. Also, it reveals a reliable strategy for achieving higher electromagnetic wave absorbing performance through improving the interfacial polarization and conductivity of the composite absorber.

## Figures and Tables

**Figure 1 nanomaterials-14-01566-f001:**
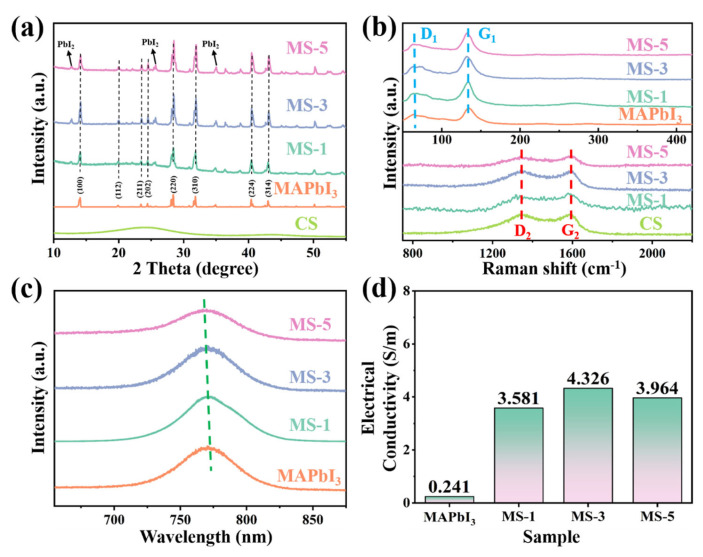
The XRD spectra (**a**), Raman spectra (**b**), PL spectra (**c**), and conductivity plot (**d**) of the MAPbI_3_/CS composite.

**Figure 2 nanomaterials-14-01566-f002:**
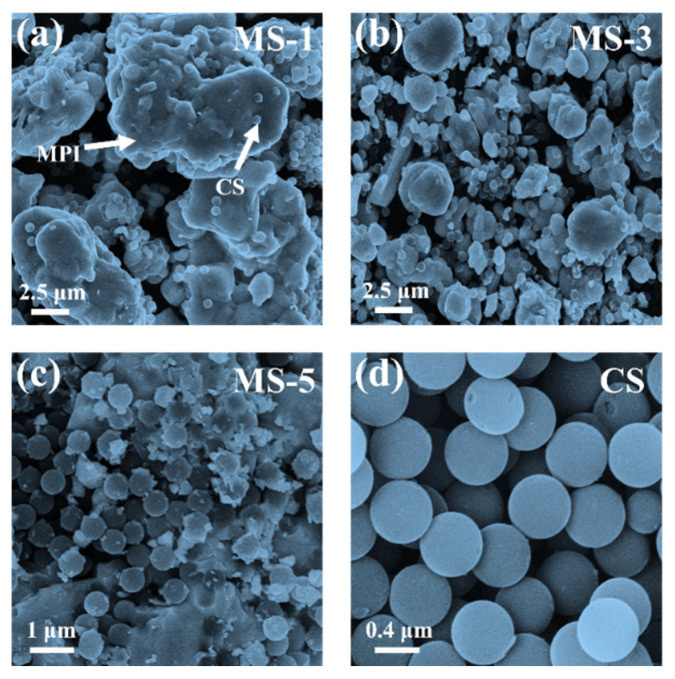
SEM photograph of MS-1 (**a**), MS-3 (**b**), MS-5 (**c**), and CS (**d**).

**Figure 3 nanomaterials-14-01566-f003:**
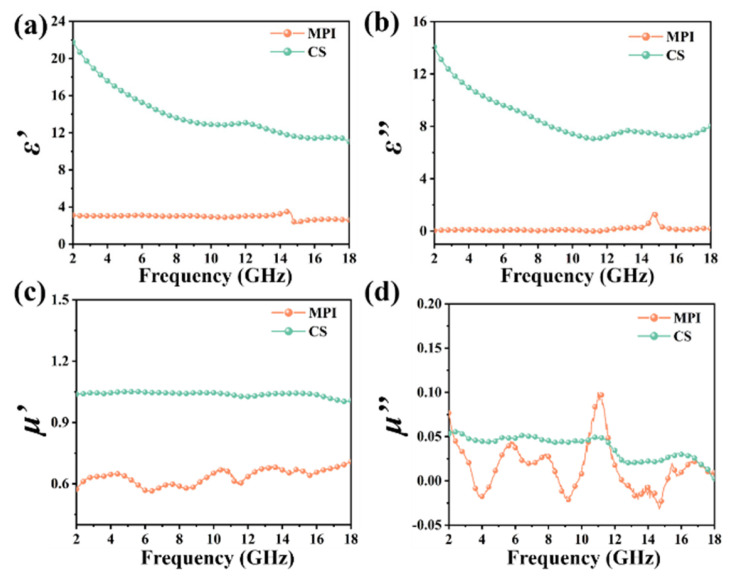
Real part of complex permittivity ε′ (**a**), imaginary part of complex permittivity ε″ (**b**), real part of complex permeability μ′ (**c**), and imaginary part of complex permeability μ″ (**d**) of MAPbI_3_ and CSs within 2–18 GHz.

**Figure 4 nanomaterials-14-01566-f004:**
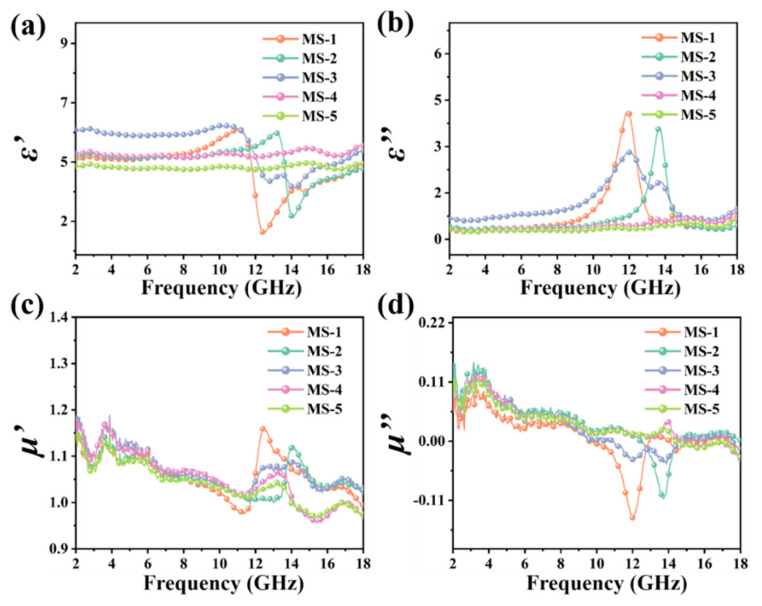
Real part of complex dielectric (ε′) (**a**), imaginary part of complex dielectric (ε″) (**b**), real part of complex permeability (μ′) (**c**), and imaginary part of complex permeability (μ″) (**d**) of MAPbI_3_/CS composites within 2–18 GHz.

**Figure 5 nanomaterials-14-01566-f005:**
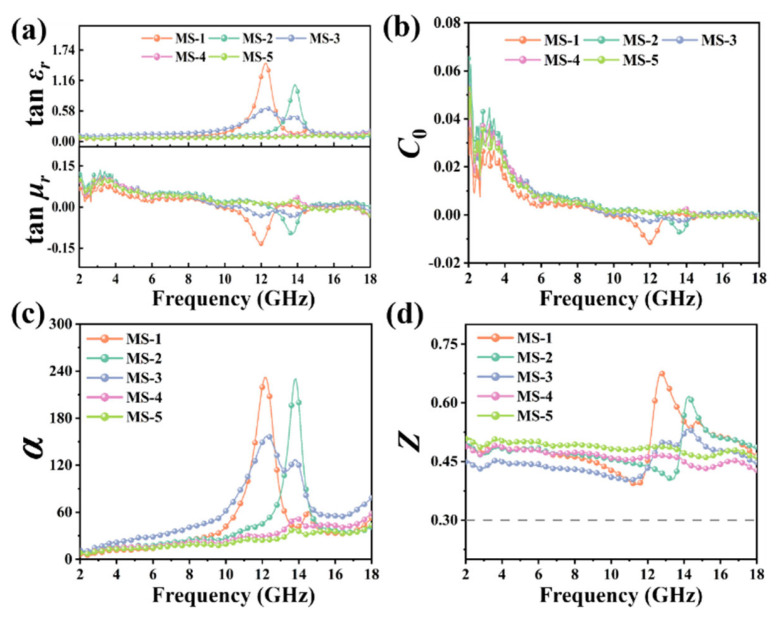
Dielectric loss tangent and magnetic loss tangent (**a**), eddy current loss constant (**b**), attenuation loss coefficient (**c**), and intrinsic impedance ratio (**d**) of MAPbI_3_/CS composites.

**Figure 6 nanomaterials-14-01566-f006:**
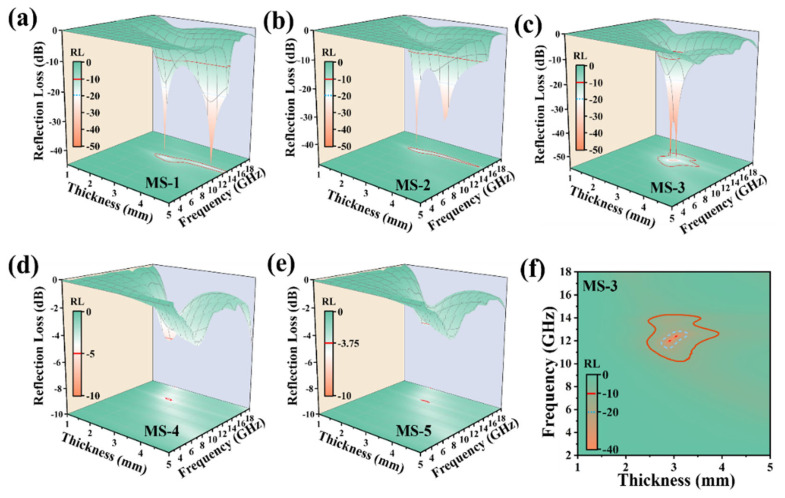
Schematic diagram of three-dimensional reflection loss performance of MS-1 (**a**), MS-2 (**b**), MS-3 (**c**), MS-4 (**d**), and MS-5 (**e**), and two-dimensional reflection loss performance of MS-3 (**f**).

**Figure 7 nanomaterials-14-01566-f007:**
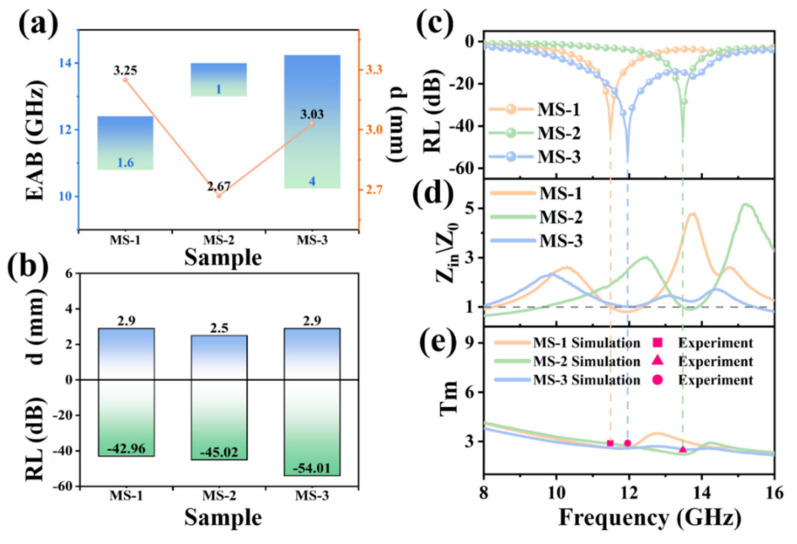
EAB properties of MAPbI_3_/CS and corresponding thicknesses (**a**), maximum reflection loss intensity of MAPbI_3_/CS and corresponding thicknesses (**b**), maximum reflection loss intensity of MAPbI_3_/CS (**c**), normalized input impedance of MAPbI_3_/CS (**d**), and matched thickness correlation comparison chart (**e**).

## Data Availability

The data that support the findings of this study are available from the corresponding authors upon reasonable request.

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
