# Peer review of "A Hybrid Perovskite-Based Electromagnetic Wave Absorber with Enhanced Conduction Loss and Interfacial Polarization through Carbon Sphere Embedding"

_nanomaterials, 2024, doi:10.3390/nano14191566_

Round 1

Reviewer 1 Report

Comments and Suggestions for Authors

This manuscript describes the preparation and electromagnetic wave absorption performance of perovskite materials and carbon spheres. The authors synthesized MAPbI3, a material widely used in perovskite solar cells, and mixed it with carbon spheres in various ratios to create composite materials. These composites were then mixed with paraffin wax, and their dielectric constant and permeability were measured to calculate the reflection loss.

Overall, this manuscript is well-written, and the analyses related to electromagnetic wave absorption are thoroughly performed. This reviewer believes that this manuscript could be suitable for publication with the following revisions:

  • In Figure 2, the images have been modified with a blue tone, which is unnecessary. It would be preferable to convert them to a standard grayscale format for publication.
  • In Figure 3, why is the real part of the permeability of MP1 smaller than 1?
  • In Figure 4, it can be observed that the dielectric constant and permeability suddenly change at the same frequency (around 12 GHz for MS-1 and around 14 GHz for MS-2). Could this issue arise from calculating permittivity and permeability from the S-parameters rather than from resonance?

Author Response

Reviewer #1:This manuscript describes the preparation and electromagnetic wave absorption performance of perovskite materials and carbon spheres. The authors synthesized MAPbI3, a material widely used in perovskite solar cells, and mixed it with carbon spheres in various ratios to create composite materials. These composites were then mixed with paraffin wax, and their dielectric constant and permeability were measured to calculate the reflection loss.

Overall, this manuscript is well-written, and the analyses related to electromagnetic wave absorption are thoroughly performed. This reviewer believes that this manuscript could be suitable for publication with the following revisions:

Comment 1. In Figure 2, the images have been modified with a blue tone, which is unnecessary. It would be preferable to convert them to a standard grayscale format for publication.

Response 1: We thank the reviewer for the helpful comment to improve our manuscript. In the revised manuscript, we convert them to a standard grayscale format.

Figure 2. SEM photograph of MS-1 (a), MS-3 (b), MS-5 (c), and CS (d).

Comment 2. In Figure 3, why is the real part of the permeability of MPI smaller than 1?

Response 2: We thank the reviewer for this very professional comment. Theoretically, for non-magnetic system, the magnetic permeability is 1 in the real part and 0 in the imaginary part, indicating that they do not have a magnetic loss mechanism. Thus, for MAPbI3, a typical dielectric material, the μ′ should also be around 1. However, in this paper, the μ′ of MAPbI3 is smaller than 1, which is very abnormal.

According the published results (J. Mater. Chem. C, 2018, 6, 9615-9623), this abnormal phenomenon may arise from the radiation of magnetic energy for magnetic materials. Under the applied electromagnetic field, the motion of charges in material will produce an ac electric field, and this electric field can further induce an internal magnetic field according to the Maxwell equations. The induced magnetic field may counteract or surpass the magnetic branch of the external electromagnetic field, resulting in the decreased magnetic loss ability, which may cause the decrease of the μ′.

Furthermore, in the published works about microwave absorption of MAPbX3 (X=I, Br, Cl) (J. Mater. Chem. C, 2018, 6, 4201-4207), the authors found a slight increase of μ′ (1.4, 1.3 and 1.5) at the higher frequencies of 16.9, 16.6 and 15.9 GHz, suggests that magnetic loss still happens despite the absence of non-magnetic properties in these perovskites.

Therefore, there are many questions that need further investigation on the microwave absorption of hybrid perovskite. We sincerely admit that we cannot explain this phenomenon in this paper and hope the reviewer or any expert could give us any advice.

Comment 3. In Figure 4, it can be observed that the dielectric constant and permeability suddenly change at the same frequency (around 12 GHz for MS-1 and around 14 GHz for MS-2). Could this issue arise from calculating permittivity and permeability from the S-parameters rather than from resonance?

Response 3: We thank the reviewer for this helpful comment. In this paper, the absorbing performance test of materials is based on network analysis method, and the scattering parameters of samples are obtained by means of vector network analyzer. Then the electromagnetic parameters of samples are calculated from S-parameters.

In the published works (J. Mater. Chem. C, 2018, 6, 4201-4207), the dielectric constant and permeability also both change at the same frequency. The peaks of dielectric constant exhibits resonance and the loss mechanisms are attributed to dominant dipolar polarization.

Furthermore, similar characteristics of permeability imply that the loss mechanism of the perovskites also includes magnetic loss. Generally, magnetic loss mainly results from magnetic hysteresis, domain-wall displacement, natural resonance and eddy current loss. If the magnetic loss results from eddy current loss, the C0 values should be constant when the frequency varies. However, the values of C0 have two strong resonance peaks as a function of frequency (Figure. 5b for our work). Thus, it can be concluded that the magnetic loss for MAPbI3 perovskites is mainly ascribed to the natural resonance, which makes the behaviors of dielectric and magnetic losses more complex.

Thus, this issue mainly arises from resonance, rather calculating permittivity and permeability from the S-parameters. 

Reviewer 2 Report

Comments and Suggestions for Authors

The manuscript is well written. However, the iThenticate report suggests 35% similarity, majority of which originates from authors’ own works. Please bring it down below 20%. There are few additional points to consider.

1.      Use consistent terminology, e.g., hours should be ‘h’.

2.      A few typos exist, e.g., ‘coundtion loss’ should be ‘conduction loss’, ‘Cs’ should be ‘CS’, etc.

3.      The XRD patterns should be indexed. Some additional peaks appear upon addition of CS to the perovskites. Have the authors attempted to identify these?

4.      The characterization data and discussion for MS-2 and MS-4 are missing in Figure 1.

Comments on the Quality of English Language

Some minor typos exist.

Author Response

Reviewer #2: The manuscript is well written. However, the iThenticate report suggests 35% similarity, majority of which originates from authors’ own works. Please bring it down below 20%. There are few additional points to consider.

Response: We thank the reviewer for the helpful comment to improve our manuscript. In the revised manuscript, we try to reduce the similarity as much as possible.

Comment 1. Use consistent terminology, e.g., hours should be ‘h’.

Response 1: We thank the reviewer for this helpful comment. In the revised manuscript, we replace all the “hours” with “h”.

Comment 2. A few typos exist, e.g., ‘coundtion loss’ should be ‘conduction loss’, ‘Cs’ should be ‘CS’, etc.

Response 2: We thank the reviewer for this helpful comment. In the revised manuscript, we correct all the typos.

Comment 3. The XRD patterns should be indexed. Some additional peaks appear upon addition of CS to the perovskites. Have the authors attempted to identify these?

Response 3: We thank the reviewer for the helpful comment to improve our manuscript. In the revised manuscript, we indexed the XRD patterns in Figure 1a.

Comment 4. The characterization data and discussion for MS-2 and MS-4 are missing in Figure 1.

Response 4: We thank the Reviewer for this insightful comment. In the XRD, Raman, PL and conductivity characterization, we select MS-1, MS-3 and MS-5 to analyze how the carbon affect the crystal quality of MAPbI3. And in fact, the characterization data and discussion of MS-1, MS-3 and MS-5 is necessary to get the conclusion. However, in order to obtain the optimal reflection loss performance, we did the RL characterization for both MS-2 and MS-4, which could give us a more precise composite component. Thanks the reviewer again for this comprehensive advice for our manuscript.
